# EDTA-Anticoagulated Whole Blood for SARS-CoV-2 Antibody Testing by Electrochemiluminescence Immunoassay (ECLIA) and Enzyme-Linked Immunosorbent Assay (ELISA)

**DOI:** 10.3390/diagnostics10080593

**Published:** 2020-08-14

**Authors:** Marc Kovac, Lorenz Risch, Sarah Thiel, Myriam Weber, Kirsten Grossmann, Nadja Wohlwend, Thomas Lung, Dorothea Hillmann, Michael Ritzler, Susanna Bigler, Francesca Ferrara, Thomas Bodmer, Konrad Egli, Mauro Imperiali, Sonja Heer, Yacir Salimi, Harald Renz, Philipp Kohler, Pietro Vernazza, Christian R. Kahlert, Matthias Paprotny, Martin Risch

**Affiliations:** 1Department of Rheumatology and Clinical Immunology/Allergology, University Hospital Bern, 3010 Bern, Switzerland; marc.kovac@insel.ch; 2 Departments of Infectious Diseases and Hospital Epidemiology, Children’s Hospital of Eastern Switzerland, 9007 St Gallen, Switzerland; lorenz.risch@risch.ch (L.R.); kirsten.grossmann@risch.ch (K.G.); nadja.wohlwend@risch.ch (N.W.); thomas.lung@risch.ch (T.L.); dorothea.hillmann@risch.ch (D.H.); michael.ritzler@risch.ch (M.R.); susanna.bigler@risch.ch (S.B.); francesca.ferrara@risch.ch (F.F.); thomas.bodmer@risch.ch (T.B.); konrad.egli@risch.ch (K.E.); mauro.imperiali@risch.ch (M.I.); yacir.salimi@risch.ch (Y.S.); 3Private University of the Principality of Liechtenstein, Dorfstrasse 24, 9495 Triesen, Liechtenstein; 4Liechtensteinisches Landesspital, Heiligkreuz, 9490 Vaduz, Liechtenstein; sarah.thiel@landesspital.li (S.T.); myriam.weber@landesspital.li (M.W.); matthias.paprotny@landesspital.li (M.P.); 5Blutspendedienst Graubünden, 7000 Chur, Switzerland; sonja.heer@blutspende-gr.ch; 6Institute of Laboratory Medicine and Pathobiochemistry, Molecular Diagnostics, University Hospital Marburg, 35043 Marburg, Germany; Harald.Renz@uk-gm.de; 7Cantonal Hospital St Gallen, Department of Infectious Diseases and Hospital Epidemiology, 9007 St Gallen, Switzerland; Philipp.Kohler@kssg.ch (P.K.); pietro.vernazza@kssg.ch (P.V.); Christian.Kahlert@kispisg.ch (C.R.K.); 8Children’s Hospital of Eastern Switzerland, Infectious Diseases and Hospital Epidemiology, 9007 St Gallen, Switzerland; 9Zentrallabor, Kantonsspital Graubünden, 7000 Chur, Switzerland

**Keywords:** antibodies, COVID-19, preanalytics, SARS-CoV-2, sensitivity, serum, specificity, whole blood

## Abstract

While lateral flow test formats can be utilized with whole blood and low sample volumes, their diagnostic characteristics are inferior to immunoassays based on chemiluminescence immunoassay (CLIA) or enzyme-linked immunosorbent assay (ELISA) technology. CLIAs and ELISAs can be automated to a high degree but commonly require larger serum or plasma volumes for sample processing. We addressed the suitability of EDTA-anticoagulated whole blood as an alternative sample material for antibody testing against SARS-CoV-2 by electro-CLIA (ECLIA; Roche, Rotkreuz, Switzerland) and ELISA (IgG and IgA; Euroimmun, Germany). Simultaneously drawn venous serum and EDTA-anticoagulated whole blood samples from 223 individuals were included. Correction of the whole blood results for hematocrit led to a good agreement with the serum results for weakly to moderately positive antibody signals. In receiver-operating characteristic curve analysis, all three assays displayed comparable diagnostic accuracy (area under the curve (AUC)) using corrected whole blood and serum (AUCs: 0.97 for ECLIA and IgG ELISA; 0.84 for IgA ELISA). In conclusion, our results suggest that the investigated assays can reliably detect antibodies against SARS-CoV-2 in hemolyzed whole blood anticoagulated with EDTA. Correction of these results for hematocrit is suggested. This study demonstrates that the automated processing of whole blood for identification of SARS-CoV-2 antibodies with common ECLIA and ELISA methods is accurate and feasible.

## 1. Introduction

The coronavirus disease 2019 (COVID-19) constitutes a recent global pandemic caused by the severe acute respiratory syndrome coronavirus 2 (SARS-CoV-2) virus. Whereas acute disease diagnosis by laboratory methods predominantly utilizes demonstration of virus replication by real-time reverse transcriptase polymerase chain reaction (RT-PCR), serologic tests are mainly employed for diagnosis of past COVID-19 infection [1,2,3,4,5,6,7,8,9]. Demonstration of specific antibodies against the SARS-CoV-2 also allows for confirmation of clinically suspected COVID-19 cases for which RT-PCR testing has been negative [1,10]. From a public health perspective, serologic tests are also employed for estimating the proportion of individuals already infected with the SARS-CoV-2 and facilitating contact tracing, as well as the surveillance or identification of individuals who are susceptible to COVID-19 infection [3]. 

Antibodies against the SARS-CoV-2 develop within 14–21 days after COVID-19 symptom onset [2]. Specific IgM or IgA can precede the development of specific IgG [2,11]. However, specific IgG can also develop along with or before the occurrence of specific IgM or IgA [2]. The specificity of antibodies against the SARS-CoV-2 is directed against several virus-specific proteins, such as the nucleocapsid (N) or spike (S) proteins [12,13]. N or S (S1 and/or S2) are the target antigens in most of the commonly employed immunoassays in clinical diagnostics [13,14,15,16]. It has been shown that the antibodies against the receptor-binding domain (RBD) of S1, as well as the N-protein, correlate closely with the virus neutralization titer [6]. RBD-binding enables the SARS-CoV-2 to infect the target cell via the angiotensin-converting enzyme 2 (ACE2) receptor [17].

Several assay types have been developed for detection of the COVID-19-specific immune response, including immune chromatographic lateral flow assays [18], enzyme-linked immunosorbent assays (ELISAs) [8,13,15,19], and chemiluminescence assays (CLIAs) [14,16]. Lateral flow assays were marketed early by a large variety of manufacturers. The diagnostic accuracy and utility of some of these products has been questioned. However, since lateral flow formats can be used with minimal sample volume (as little as 10 µL) and a variety of materials (e.g., whole blood, serum, and plasma), capillary blood sampling is frequently employed with this assay format [20]. Other assay formats (ELISAs and CLIAs) have been reported to possess better diagnostic characteristics and more efficient and higher throughput. However, due to the dead volume needed for trouble-free automated sample processing, these assays need more sample volume (minimum of 100–200 µL) than lateral flow tests. Further, these assays have only been validated for cell-free matrices, such as serum and plasma.

Since the frequency of requests for the COVID-19 antibody test continues to increase, automated testing is becoming more and more important. As capillary samples are characterized by low sample volume, and separation of cells from plasma or serum further reduces the sample volume [21], capillary samples are rarely processed on high-throughput laboratory analyzers [22]. The issue of low sample volume associated with capillary blood sampling for testing with CLIAs and ELISAs may be circumvented by employing whole blood as a matrix. A total sample volume of 200 µL whole blood would allow for automated and efficient processing of capillary blood samples for SARS-CoV-2 antibody testing with CLIAs or ELISAs. Thus, we investigated whether the results from SARS-CoV-2 antibody testing with EDTA-anticoagulated whole blood would be comparable to the results obtained in serum.

## 2. Materials and Methods

### 2.1. Study Setting and Study Population

In this study, we analyzed paired serum and EDTA-whole blood samples from individuals who were tested for SARS-CoV-2 antibodies. The study was conducted with anonymized blood samples from patients referred to the labormedizinisches zentrum Dr. Risch Ostschweiz AG (Buchs SG, Switzerland). The study protocol was verified by the ethics board of Eastern Switzerland (EKOS; BASEC Nr. Req-2020-00586; approval date 11.5.2020), which waived informed consent for performing laboratory analysis on anonymized samples. Samples from patients referred for COVID-19 serology for whom an EDTA-whole blood sample was available from the same venipuncture were included.

### 2.2. Data Collection and Measurements

For each sample, the age and gender of the individual, the results from SARS-CoV-2 RT-PCR analysis (if performed), and the delay from RT-PCR analysis to antibody testing (days) were documented. Serum and EDTA-whole blood samples referred for complete blood count testing were kept at 4 °C to 8 °C before analysis of SARS-CoV-2 antibodies, which was conducted daily on weekdays. Before placing the serum samples on laboratory analyzers, the samples were brought to room temperature over 4 h and homogenized by vortexing. Before antibodies were identified in EDTA-whole blood, hemolysis was induced by freezing the sample to −80 °C for 2 h after homogenization on a sample roller for 30 min. After freezing, the hemolyzed EDTA-whole blood samples were brought to room temperature over 4 h on a sample roller and then put on the analyzer. For antibody testing using ECLIA, the antibodies were tested on a COBAS 6000 (Roche Diagnostics, Rotkreuz, Switzerland) using the Elecsys^®^ Anti-SARS-CoV-2 assay (Roche Diagnostics, Rotkreuz, Switzerland). This assay was CE-marked, as well as cleared by the U.S. Food and Drug Administration (FDA), and employs a recombinantly engineered nucleocapsid antigen for detection of total immunoglobulin. According to the manufacturer, serum and EDTA-plasma or heparin-plasma, but not EDTA-whole blood have been specified as acceptable specimens. Additionally, the samples were analyzed by ELISA (EUROIMMUN Anti-SARS-CoV-2 IgG and IgA ELISAs, Euroimmun, Luzern, Switzerland) run on a DSX analyzer (Dynex Technologies, Denkendorf, Germany) 15. These CE-marked assays detect specific IgG and IgA directed against the S1 spike protein of the SARS-CoV-2. Serum, EDTA-plasma, heparin-plasma, and citrate-plasma, but not EDTA-whole blood have been specified by the manufacturer as acceptable specimens. Hematocrit was assessed before inducing hemolysis in the EDTA-whole blood samples using a Sysmex XS-1000i instrument (Sysmex, Horgen, Switzerland), employing the cumulative pulse height method 20. The inter-series coefficients of variation (CVs) were 7.1% for the Elecsys^®^ Anti-SARS-CoV-2 assay (at a mean cutoff index [COI] of 26.6; COI for positivity > 1), 7.8% for the EUROIMMUN Anti-SARS-CoV-2 IgG ELISA (mean ratio of the extinction of the control patient sample over the extinction of the calibrator, [S/C] of 2.67; S/C for positivity > 1.1), 8.6% for the EUROIMMUN Anti-SARS-CoV-2 IgA ELISA (mean S/C of 2.54; S/C for positivity > 1.1), and 0.4% for hematocrit (at a mean level of 35.4%).

### 2.3. Statistical Methods

Continuous variables are presented as medians and interquartile ranges (IQRs), whereas proportions are given as percentages with 95% confidence intervals (CIs). Associations between variables were calculated with the Spearman’s rank correlation. Proportions were compared using the chi-square tests. Results obtained by the ECLIA assay, as well as the ELISAs measuring EDTA-whole blood were corrected for hematocrit by multiplying the result with the reciprocal of the hematocrit to render them comparable to the serum measurements. The Bland–Altman analysis was performed by comparing the results corrected for measured hematocrit to the results obtained in serum. This type of analysis assesses accuracy and precision of the antibody measurements by relating the difference between the hemolyzed EDTA-whole blood and serum measurements to the average of the two results in each patient. The limits of agreement are given by the mean +1.96 standard deviations (SDs) containing 95% of the values. The mean difference is a measure for accuracy, and the limits of agreement are a measure of precision. A method comparison was also performed using the Passing–Bablok regression analysis. The proportion of positive results in the EDTA-whole blood samples among the positive results obtained by the same method in the corresponding serum samples, as well as the proportion of negative results in the EDTA-whole blood samples among the negative results in serum were calculated. Diagnostic sensitivities and specificities for the cutoffs provided by the manufacturers were calculated for serum, EDTA-whole blood samples without hematocrit correction, and EDTA-whole blood results corrected for hematocrit. Further, diagnostic accuracy was assessed for the different sample materials and assays by means of receiver-operating characteristic (ROC) curves, which were compared by the method of DeLong 21. As a reference standard for determination of disease status, a positive RT-PCR result or one N- along with one S-antigen-positive antibody assay was considered to be COVID-19-positive, whereas cases with no or negative PCR and negative results from one N- and one S-antigen assay were considered negative for COVID-19. Finally, *p*-values < 0.05 were considered to be statistically significant. The Medcalc software version 18.11.3 (Mariakerke, Belgium) was used for all statistical calculations.

## 3. Results

### 3.1. Baseline Characteristics

Paired serum and EDTA-whole blood samples were available from 223 patients (112 females, 50%; 95% CI: 44, 57). The median age of the patients was 40 years old (IQR: 27, 55). COVID-19 was diagnosed in 110 patients (49%; 95% CI: 43, 56) with either RT-PCR (n = 72) or two positive antibody results in one assay against the N-antigen and one assay against the S-antigen (IgG or IgA; n = 38). For all patients with RT-PCR results (n = 133), serum samples were taken after a median of 50 days (IQR: 43, 54) following RT-PCR. Serum antibody testing was positive with the ECLIA, IgG ELISA, and IgA ELISA in 107, 99, and 95 samples, respectively. The median hematocrit was 42% (IQR: 40, 45).

### 3.2. Association between Serum and Whole Blood Results

We observed a close correlation between the results obtained in serum and hemolyzed EDTA-whole blood with or without correcting for hematocrit. The correlations between the results obtained in serum and hemolyzed EDTA-whole blood not corrected for hematocrit were r = 0.92 (*p* < 0.001) in the IgG ELISA, r = 0.87 (*p* < 0.001) in the IgA ELISA, and r = 0.9 (*p* < 0.001) in the ECLIA. The respective correlations between the serum and hemolyzed EDTA-whole blood results corrected for hematocrit were similar: r = 0.89 (*p* < 0.001) in the IgG ELISA, r = 0.86 (*p* < 0.001) in the IgA ELISA, and r = 0.89 (*p* < 0.001) in the ECLIA.

The Passing–Bablok regression analysis with the serum results as the independent variable and the whole blood results not corrected for hematocrit as the dependent variable revealed a slope (s) of 0.67 and an intercept (i) of 0.01 for the IgG ELISA, 0.55 (s) and 0.02 (i) for the IgA ELISA, and 0.53 (s) and −0.05 (i) for the ECLIA. The respective parameters for serum and whole blood corrected for hematocrit were 1.57 (s) and −0.04 (i) for the IgG ELISA, 1.36 (s) and −0.04 (i) for the IgA ELISA, and 1,31 (s) and −0.13 (i) for the ECLIA. Inspection of the Bland–Altman plots (Figure 1) illustrated that the results between serum and whole blood corrected for hematocrit were comparable up to a S/C of 2 in the IgG ELISA (cutoff for positivity S/C ≥ 1.1; Figure 1a) and 3 in the IgA ELISA (cutoff for positivity S/C ≥ 1.1; Figure 1b), as well as up to a COI of 25 in the ECLIA (cutoff for positivity COI ≥ 1; Figure 1c). It can thus be concluded that the anti-SARS-CoV-2 antibody results in whole blood corrected for hematocrit with weakly and moderately positive findings are comparable to those obtained from serum.

### 3.3. Detection Rates of Whole Blood Compared to Serum

In the whole blood results not corrected for hematocrit, 101 of the 107 positive ECLIA serum results were found to be positive (94%; 95% CI: 88, 97). The other respective proportions were 81 out of 99 (82%; 95% CI: 73, 88) in the IgG ELISA and 66 out of 95 (70%; 95% CI: 60, 78) in the IgA ELISA. In the whole blood results corrected for hematocrit, 106 of the 107 positive ECLIA serum results were found to be positive (99%; 95% CI: 95, 99.8) along with 96 out of 99 (97%; 95% CI: 92, 99) in the IgG ELISA and 90 out of 95 (95%; 95% CI: 88, 98) in the IgA ELISA.

In the whole blood results not corrected for hematocrit, 115 of the 116 negative ECLIA serum results were found to be negative (99%; 95% CI: 95, 99.8) in addition to 122 out of 124 (98%; 95% CI: 94, 99.5) results in the IgG ELISA and 127 out of 128 results (99%; 95% CI: 96, 99.8) in the IgA ELISA. In the whole blood results corrected for hematocrit, 115 of the 116 negative ECLIA serum results were found to be negative (99%; 95% CI: 95, 99.8). The other respective proportions were 115 out of 124 (93%; 95% CI: 87, 96) results in the IgG ELISA and 114 out of 128 (89%; 95% CI: 83, 93) results in the IgA ELISA.

### 3.4. Evaluation of Hematocrit Correction

Among the initially negative ECLIA results in serum, one sample showed a positive result in the whole blood results corrected for hematocrit (COI in serum/corrected whole blood: 0.1/3.44). This sample had positive results in the IgG and IgA ELISAs. Thus, this sample can be considered as a false-negative in the ECLIA serum results.

Among the initially negative IgG ELISA results in serum, nine samples showed positive results in the whole blood results corrected for hematocrit (S/C in serum/corrected whole blood: 1/1.5; 1/1.63; 0.6/1.25; 0.7/1.25; 1/1.14; 0.1/3.26; 0.3/4.88; 0.9/2.63; 0.9/1.16). Interestingly, all but one sample had positive ECLIA results, and the remaining sample (0.9/1.16) had a positive IgA ELISA result (S/C: 3.7). It could thus be concluded that all nine samples were likely false-negatives in the IgG ELISA serum results. Therefore, the specificity of the IgG ELISA assay using whole blood corrected for hematocrit is unaffected compared to its serum performance.

Among the initially negative IgA ELISA results in serum, 14 samples showed positive results in the whole blood results corrected for hematocrit (S/C in serum/corrected whole blood: 0.3/1.25; 0.8/1.25; 0.8/2.14; 1/1.22; 1/1.52; 0.6/1.43; 0.9/1.16; 0.3/1.19; 0.9/1.5; 0.2/10.98; 0.3/1.43; 1/1.4;1/1.35; 1/1.28). Only six of these 14 samples had positive ECLIA results. The other samples were also clearly negative in the IgG ELISA. It can thus be concluded that the specificity of the IgA ELISA using whole blood corrected for hematocrit decreases compared to its serum performance.

In summary, correcting the whole blood results for hematocrit improved the sensitivity of the whole blood measurement, while leaving specificity relatively unchanged compared to the ECLIA serum results. Correcting the whole blood results for hematocrit in the IgG ELISA increased the sensitivity of the method compared to serum assessments, while specificity remained unchanged. Finally, correcting the whole blood results for hematocrit in the IgA ELISA improved sensitivity at the cost of a somewhat decreased specificity. Given these findings, it can be concluded that for the ECLIA and IgG ELISA, the results are not inferior when using whole blood corrected for hematocrit instead of serum.

### 3.5. Diagnostic Sensitivity and Specificity of Serum and Whole Blood

The diagnostic sensitivities and specificities of the different assays are shown in Table 1.

Using the predefined cutoffs provided by the manufacturers, the ECLIA in serum exhibited a significantly higher diagnostic sensitivity than both ELISAs (*p* < 0.01 for both). In whole blood corrected for hematocrit, the sensitivities of the ECLIA and IgG ELISA did not differ significantly. The IgG ELISA had a significantly higher diagnostic sensitivity than the IgA ELISA (*p* < 0.05). In serum and whole blood corrected for hematocrit, the diagnostic specificities were significantly better for the ECLIA and IgG ELISA than the IgA ELISA (*p* < 0.05), whereas there was no significant difference between the ECLIA and IgG ELISA. For all three of the investigated assays, changes in the diagnostic sensitivities and specificities were not significantly different when using whole blood corrected for hematocrit instead of serum.

### 3.6. Diagnostic Accuracy of Whole Blood Assays

ROC analysis revealed comparable AUCs between the different materials for all of the investigated assays: 0.987 for serum, 0.973 for whole blood, and 0.973 for whole blood corrected for hematocrit in the ECLIA; 0.969, 0.971, and 0.971 in the IgG ELISA; and 0.939, 0.935, and 0.936 in the IgA ELISA. There were no statistical differences in the AUCs between the ECLIA and IgG ELISA regardless of the sample material used. Both assays for all materials were significantly better than the IgA ELISA using whole blood with or without correction for hematocrit (*p* < 0.05). The AUCs of the different assays in serum and hemolyzed EDTA-whole blood with correction for hematocrit are shown in Figure 2.

## 4. Discussion

The three assays had comparable diagnostic characteristics when using serum or hemolyzed EDTA-whole blood with or without correction for hematocrit for analysis. With conventional cutoffs, correction of the whole blood results for hematocrit appeared to preserve, or even increase sensitivity in all methods. Together, our results support the use of whole blood as a valid material for anti-SARS-CoV-2 antibody testing with the three investigated assays.

To the best of our knowledge, the only anti-SARS-CoV2 tests that have been approved for EDTA-whole blood use are immunochromatographic lateral flow rapid tests [20]. Even though some of these formats have acceptable diagnostic characteristics, especially for specific IgG-antibodies, lateral flow test formats are considered inferior to commonly used automated assays employing CLIA or ELISA techniques [18]. Automated immunoassays, however, have not been validated with whole blood. Our data show that whole blood constitutes an acceptable sample material for automated testing in the investigated test formats. This finding is important because it allows for the use of capillary blood on automated analyzers. Such analyzers have a dead volume of up to 200 µL, and COVID-19 tests need test volumes that are approximately 20 µL. Collection of capillary blood is a method that can be used outside of medical institutions, which may present an advantage when testing is performed on a population level. In our experience, capillary samples rarely exceed a volume of 200 µL. Centrifuging such samples in order to obtain plasma or serum is laborious and does not provide enough sample volume (< 100 µL) for appropriate processing. Utilizing the entire blood sample of 100–200 µL without centrifugation may allow for the use of capillary blood samples in highly automated CLIA or ELISA analysis systems. Until now, capillary blood samples were primarily utilized for lateral flow tests. In addition to a somewhat inferior performance, these tests are limited by the fact that they cannot be automatized and do not allow for the provision of numeric results to assess antibody response development. Thus, the combination of capillary blood sampling and high-quality tests may be very important in terms of planning and conducting large epidemiological studies [23].

Our data show that correcting for hematocrit aids detection rates in all three investigated assays compared to the results of whole blood without correction. Should it not be possible to obtain a hematocrit measurement, the whole blood results may be corrected for a mean hematocrit value (e.g., 42%) to be used with the conventional antibody cutoffs. The sensitivity and specificity of such an approach reveal comparable results to those provided for whole blood corrected for measured hematocrit in Table 1 (data not shown). However, it would also run the risk of false-positive antibody results, especially in patients with anemia, whereas false-negative antibody results would be found in patients with polyglobulia. In order to prevent misclassification as much as possible, we recommend correcting for hematocrit. If this is not possible, a COI value of < 0.2 in the ECLIA or a S/C value < 0.22 in the IgG and IgA ELISAs regardless of the hematocrit result would, with a high probability, correspond to a negative result in serum, even in patients with anemia (not lower than a hematocrit of 20%, which corresponds to a hemoglobin level of 67 g/L using the rule of three for converting hematocrit into hemoglobin levels [24]).

We did not assess the effect of EDTA concentration on antibody results, such as in the case of inadequately filled tubes [25]. Increased EDTA concentration leads to increased binding of metallic ions (e.g., europium, zinc, and magnesium), which are either used as immunoassay reagents (europium) or as cofactors (zinc and magnesium) for the enzymes used in signal generation, such as alkaline phosphatase [26]. Further, it has been shown that reagent antibodies in immunoassays can recognize divalent cation complex binding sites on proteins [27]. Reduced availability of such cation complexes may induce conformational changes in proteins with altered immunoreactivity [27]. We do not believe that such interference would affect the investigated assays because we do not expect conformational changes in the epitope-recognizing site in the analyte (SARS-CoV-2 antibodies). Additionally, neither europium nor alkaline phosphatase is employed as a reporter system in the investigated assays. A further limitation of the present study is that only one chemiluminescence assay, one IgG ELISA, and one IgA ELISA format have been employed. It is therefore not possible to infer that our findings would be applicable to all other ELISA and chemiluminescence assay formats. However, we do not believe that these limitations invalidate our findings.

## 5. Conclusions

In conclusion, we demonstrated that EDTA-anticoagulated whole blood represented a sample material that could be employed with commonly used immunoassay formats that allowed for highly automated throughput of samples for SARS-CoV-2 antibody testing. In clinical practice, serum samples are not always available for analysis, and our investigation may help patients with a need for SARS-CoV-2 antibody testing who are in this situation. The fact that whole blood was successfully utilized in the investigated test formats suggests that capillary blood samples, if properly taken, might also be suitable for SARS-CoV-2 antibody testing—not only with lateral flow test formats, but also immunoassays of higher quality. Capillary blood samples have already been shown to facilitate epidemiological studies of infectious disease antibodies by means of home sampling [23]. Thus, the present study is useful for validating the aforementioned conditions for epidemiological studies or clinical practice.

## Figures and Tables

**Figure 1 diagnostics-10-00593-f001:**
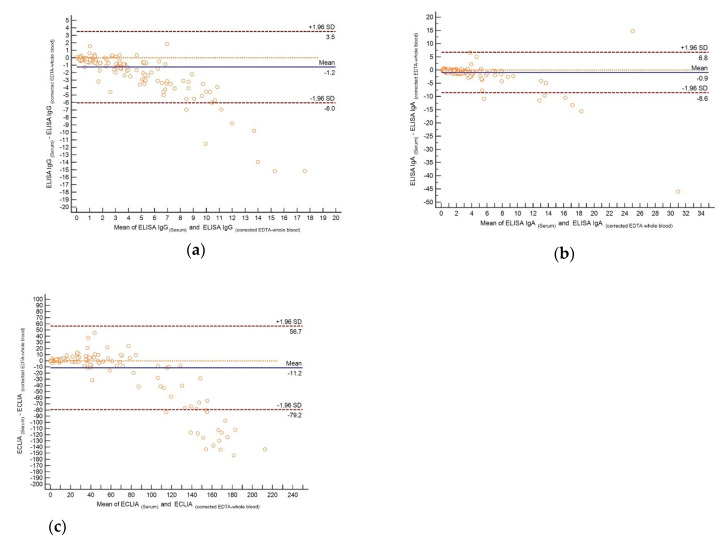
The Bland–Altman plot for assessing comparability of the results obtained in serum and EDTA-whole blood corrected for hematocrit in different assays: (**a**) IgG ELISA, (**b**) IgA ELISA, and (**c**) ECLIA.

**Figure 2 diagnostics-10-00593-f002:**
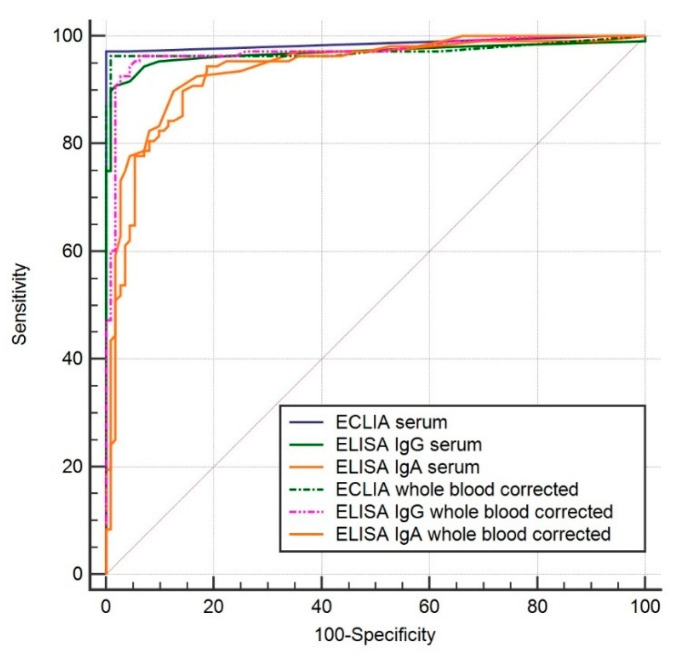
Diagnostic accuracies of the three different assays for COVID-19 identification in serum and whole blood corrected for hematocrit.

**Table 1 diagnostics-10-00593-t001:** Diagnostic sensitivities and specificities of the IgG and IgA ELISAs, as well as the ECLIA for COVID-19 diagnosis. Cutoff for the ELISAs: S/C > 1.1. Cutoff for the ECLIA: COI > 1.

	Serum		Whole Blood Corrected for Hematocrit	
	Sensitivity% [95% CI]	Specificity% [95% CI]	Sensitivity% [95% CI]	Specificity% [95% CI]
**IgG ELISA**	88% [80, 93]	99% [95, 99.8]	93% [86, 96]	97% [93, 99]
	(97/110)	(112/113)	(102/110)	(110/113)
**IgA ELISA**	78% [70, 85]	93% [87, 96]	84% [76, 89]	89% [81, 93]
	(86/110)	(105/113)	(92/110)	(100/113)
**ECLIA**	97% [92, 99]	100% [97, 100]	96% [91, 99]	99% [95, 99.8]
	(107/110)	(113/113)	(106/110)	(112/113)

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
