# Peer review of "EDTA-Anticoagulated Whole Blood for SARS-CoV-2 Antibody Testing by Electrochemiluminescence Immunoassay (ECLIA) and Enzyme-Linked Immunosorbent Assay (ELISA)"

_diagnostics, 2020, doi:10.3390/diagnostics10080593_

Round 1

Reviewer 1 Report

The authors in this manuscript have compared  the use of whole blood and serum in diagnostic testing for Sars Cov2 using 3 different assays. The authors suggest that  correcting for hematocrit aids detection rates in all three investigated assays.

Using whole blood for ELISA mass testing of viruses has been reported before. Normalizing with hematocrit is  has limitations and the authors have mentioned this in their discussion. It would be more convincing if the study had shown how hematocrit influenced the measurements by artificially altering hematocrits in the lab.

Reviewer 2 Report

The need for reliable, efficient and quick testing methods for SARS-Co V-2 antibody is increasing. The existing methods involve high throughput laboratory analyzers, and these methods are limited by the requirement of high sample volume. Capillary samples are easy to collect and can be the best sample type when testing regularly at population level. But since capillary samples are characterized by low sample volume, and the high-throughput methods also need separation of blood cells and plasma, capillary samples are rarely used for CLIA and ELISA based SARS-Co V-2 antibody testing.

The authors of this manuscript have investigated whether they can use EDTA-anticoagulated whole blood for SARS-CoV-2 antibody testing with CLIAs or ELISAs and have found that a total sample volume of 200µl whole blood would allow for automated and efficient processing of capillary blood samples, which can be used for SARS-CoV-2 antibody testing with CLIAs or ELISAs. Diagnostic accuracy of whole blood assays ROC analysis revealed comparable result with that of serum. This finding is important because it allows for the use of capillary blood on automated analyzers, and can present an advantage if testing is performed on a population level.